# Temperature Prediction of a Temperature-Controlled Container with Cold Energy Storage System Based on Long Short-Term Memory Neural Network

**Jiaming Guo [1,2,3], Dongfeng Liu [1,2], Shitao Lin [1,2], Jicheng Lin [1,2] and Wenbin Zhen [1,2,*]**

[1]  College of Engineering, South China Agricultural University, Guangzhou 510642, China; jmguo@scau.edu.cn (J.G.)
[2]  Key Laboratory of Key Technology of Southern Agricultural Machinery and Equipment, Ministry of Education, South China Agricultural University, Guangzhou 510642, China
[3]  Maoming Branch Center of Lingnan Modern Agricultural Science and Technology Guangdong Laboratory, Maoming 525000, China
[*]  Correspondence: wenbinzhen@scau.edu.cn

**Abstract:** Temperature prediction is important for controlling the environment in the preservation of fresh products. The phase change materials for cold storage make the heat transfer process complex, and the use of physical models for characterization and temperature prediction can be challenging. In order to predict the variation of the thermal environment in a temperature-controlled container with a cold energy storage system, we propose an LSTM model based on historical temperature data in which the trends of temperature variations of the fresh-keeping area, the phase change material (PCM), and the fresh products can be predicted immediately without considering the complex heat transfer process. An experimental platform of a temperature-controlled container with a cold energy storage system is built to obtain the experimental data for the prediction model's construction and validation. The prediction results based on the LSTM model are compared to the results of a physical model. In order to optimize the input data for better prediction performance, the proportion of input samples from the dataset is set to 80%, 50%, 20%, and 10%. The prediction results from different input groups are compared and analyzed. The results show that the LSTM model is able to accurately predict temperature variations of the fresh-keeping area and products, and the predicted values are in agreement with the actual values. The LSTM-based prediction model has a higher accuracy compared to the physical-based prediction model; the RMSE, MAE, and MAPE are 0.105, 0.103, and 0.010, respectively, and the relative error for the prediction of effective control hours of environmental temperature is 0.92%. It is suggested to use the initial 20% of the historical temperature data as the input to predict the future temperature variation for better prediction performance. The results of this paper offer valuable insights for accurate temperature prediction in the fresh-keeping environment with a cold energy storage system.

**Keywords:** temperature prediction; LSTM neural network; phase change materials; fresh products

## 1. Introduction

Cryogenic storage and transportation are considered the most effective methods for maintaining the freshness of fruits and vegetables through temperature control [1]. However, as energy deficiency and carbon emission regulations become increasingly stringent [2], the running cost of the cryogenic storage and transportation facility is rapidly growing [3]. Cold storage technology allows for the storage of cold in phase change material (PCM) at midnight [4,5], taking advantage of lower-priced electric power [6]. This facilitates energy shifting, helping to fill in demand valleys, reduce peak load, and lower grid costs. Thus, cold storage technology can be used to improve the temperature control of cryogenic storage and transportation facilities; it can not only reduce the running cost but

also help maintain quality by reducing temperature fluctuations in the storage or transport process, effectively reducing the loss of fresh products [4].

Temperature control is crucial for maintaining the quality of fruits and vegetables during storage and transportation [7]. Accurate temperature prediction can enhance the operation of intelligent temperature control systems [8], thereby improving system efficiency. Numerous studies are addressing this issue. Laguerre et al. predicted temperature variations at different locations of a phase change refrigeration holding tank by developing a one-dimensional analytical model [9–11]. Choi et al. developed a packaging structure thermal resistance calculation model for predicting refrigerant mass calculations [12,13].

Temperature prediction in cold energy storage facilities is challenging because the thermal characteristics of the PCM are complex during the cold energy release process, which is also coupled with the ambient environment and the products [14]. On the other hand, describing the heat transfer process and making temperature predictions for a cold energy storage system through physical modeling can be difficult to achieve due to the phase transition and complex heat transfer process of PCM.

An intelligent prediction method can compensate for the limitations of mathematical methods, which often struggle with parameter fitting due to the complexity of the mechanisms [15]. Moreover, using neural network models to predict the temperature of complex heat transfer systems can simplify the prediction process and obtain temperature prediction results without considering the specific heat transfer behavior of complex systems.

In recent years, neural network methods have been widely used in research on temperature variation prediction. Peter et al. [16] predicted the thermal efficiency of a novel straight-through evacuated tube collector using a neural network model; several artificial neural network techniques were proposed to predict the thermal performance of an all-glass, straight-through evacuated tube solar collector. Their research shows that the relative error between the radial basis function (RBF) model and the actual value is the smallest, followed by the propagation (BP) model and support vector regression (SVR) model, with $R^2$ values of 0.9658, 0.9059, and 0.8447, in that order, which also means that these models can match the experimental data well. Moon et al. [17] developed an artificial neural network (ANN) model to determine the required time for increasing the current indoor temperature to the setback temperature. In addition, the effect of different parameters on the prediction performance was discussed. The optimal number of hidden layers, the optimal number of hidden neurons, the learning rate, and the moments were 1, 7, 0.6, and 0.7, respectively, and the $R^2$ of the prediction using the ANN model was more than 0.9 after parameter optimization.

Long short-term memory (LSTM) [18], proposed by Hochreiter and Schmidhuber in 1997, is a type of recurrent neural network (RNN) composed of memory cells and control units. It is designed to predict future data changes by extracting historical features from time-series data, making it well suited for time-series data processing. Chen et al. [19] constructed an LSTM-based deep learning network to predict the temperature state of the integrated RF module. They fine-tuned the parameters to analyze the error curves between the predicted and observed values. In this research, the effects of the initial learning rate and max epochs on the training time and prediction accuracy are discussed. When the initial learning rate is 0.0001 and the max epochs is 60, the training time of the model is 3 min and 7 s, but the error does not tend to converge until after 50 epochs of training, and the final predicted value differs significantly from the actual value, with an RMSE of 0.00225. When the learning rate is increased to 0.0005 and the max epochs is 120, the training time of the model exceeds the original for 6 min and 34 s. However, the error converges sufficiently, and the accuracy of the predicted value can reach up to 98.7%, which also indicates that the initial learning rate and max epochs are important elements affecting the convergence of the error, the training time, and the prediction accuracy. Lei et al. [20] built a self-supervised deep long short-term memory (SSDLSTM) network for real-time monitoring of pot temperature in electrolysis production in the aluminum processing industry with high accuracy and robustness. In addition, the prediction accuracy and training time of

this model are more competitive than traditional LSTM models and other RNN models. Xu et al. [21] built a model based on an LSTM deep learning network to predict the indoor temperatures of a building. Their results show that the modified model of LSTM is able to maintain the highest accuracy in both single time-step-ahead prediction and multiple time-step-ahead prediction compared to BPNN, SVW and DT, which also means that the modified LSTM model is more capable of reducing the relative error in the prediction of indoor temperatures than the traditional neural network model. Hoang et al. [22] developed four different types of LSTM models to predict the change in the temperature demand of air and fresh products when applying the demand response in a cold room. Their study shows that using a small amount of data can also maintain a low relative error in temperature prediction as well as reduce the time needed for training if the temporal characteristics of the dataset are regular enough. Meanwhile, the convolutional LSTM model and bidirectional LSTM model are more susceptible to noise than the traditional LSTM model and stacked LSTM model, which means that the reasonable location of the data collection will be important for the improvement of their prediction accuracy. The use of LSTM models in all the above studies has relatively promising results, and it can be used to predict the temperature variation in the temperature-controlled container with a cold energy storage system [23].

In summary, LSTM neural networks have been widely used for temperature prediction in the complex thermal environment. In this paper, a temperature prediction model based on an LSTM neural network is developed to predict and analyze the temperature variation in the temperature-controlled container. The results of this paper will be helpful for temperature control in the fresh-keeping environment with a cold energy storage system.

## 2. Materials and Methods

The temperature-controlled container with a cold energy storage system developed by South China Agricultural University (Guangzhou, China) is shown in Figure 1, and the specific dimensions of the container are shown in Table 1. The container is mainly composed of a cold storage area, fresh-keeping area, circulating duct, and fan. The construction material of the container is mainly glass fiber reinforced plastic (GFRP). The insulation layer consists of vacuum insulation board (VIP) and polyurethane (PU), and the properties of the materials are shown in Table 2. The airflow in the container is driven by the fans (0.13 m × 0.13 m, DC, 24 V, 0.7 A), and for the adjustment of the control system, the airflow speed when the fan is activated is standardized to 4.4 m/s. Due to the airflow circuit between the cold storage area and the freshness preservation area, the cold energy released from the PCM is continuously transported to the fresh-keeping area in order to maintain a low-temperature environment of 2–8 °C.

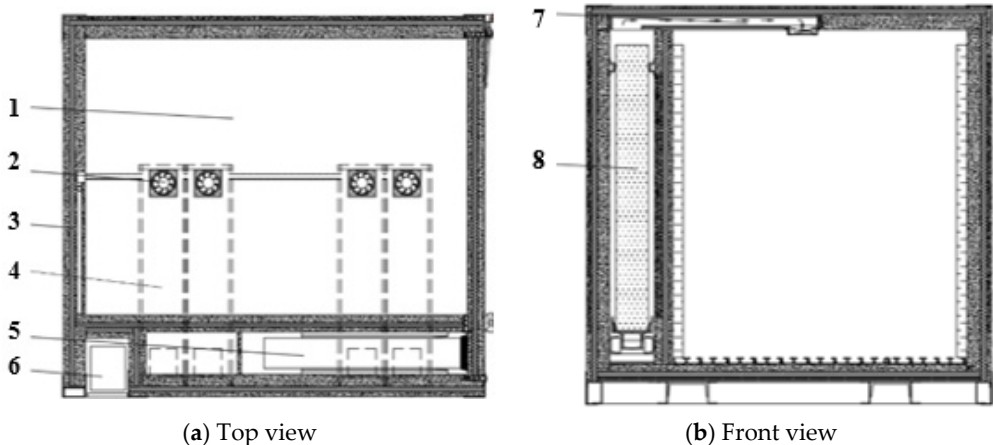

(**a**) Top view             (**b**) Front view

**Figure 1.** Schematic structure of a temperature-controlled container with a cold energy storage system: 1—fresh-keeping area; 2—fan; 3—insulation surface; 4—circulating duct; 5—cold storage area; 6—control area; 7—circulating duct; 8—cold storage plate.

**Table 1.** Specific dimensions of the container.

| Functional Zone | Lengths/mm | Widths/mm | Heights/mm |
|---|---|---|---|
| Outer dimensions of the container | 2000 | 1800 | 1720 |
| Cold storage area | 1535 | 305 | 1480 |
| Fresh-keeping area | 1800 | 1200 | 1460 |
| Dimensions within the control area | 300 | 200 | 1480 |

**Table 2.** Physical properties of materials.

| Materials | Heat Conduction /W (m·°C)$^{-1}$ | Thermal Resistance /W$^{-1}$ (m$^2$·°C) | Densities /(kg·m$^{-3}$) |
|---|---|---|---|
| GFRP | 0.4651 | 0.0086 | 2800 |
| VIP | 0.0244 | - | 45 |
| PU | 0.0048 | - | 280 |

A prototype based on the temperature-controlled container with a cold energy storage system was built and is shown in Figure 2 with a basket of fresh products, a paperless recorder, and 14 PT100 sensors.

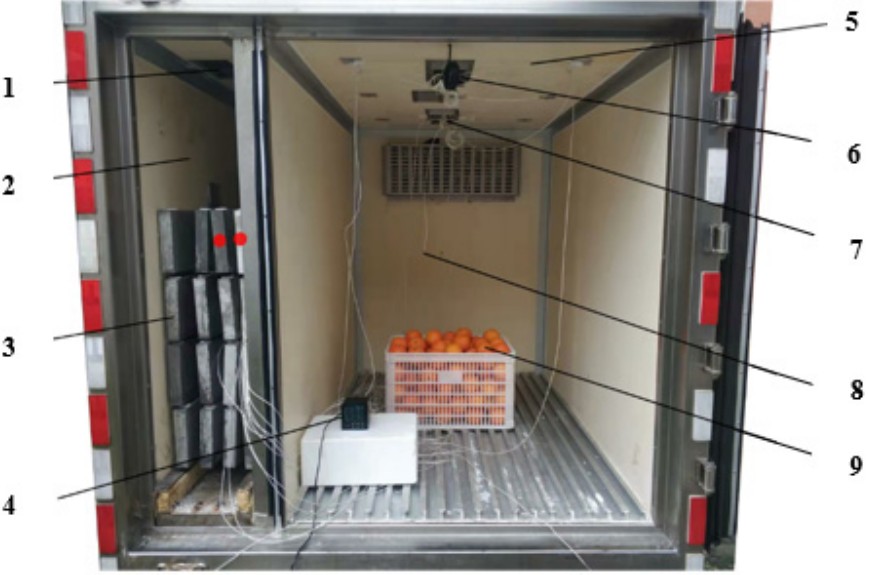

**Figure 2.** Temperature-controlled container with a cold energy storage system: 1—circulating duct; 2—cold storage area; 3—cold storage plate; 4—paperless recorder; 5—fresh-keeping area; 6—fan (outlet); 7—fan (inlet); 8—PT100; 9—navel oranges.

The temperatures of the air inlet and outlet, the center and the surface of the cold storage plate, the fresh-keeping area, the external environment, and the fresh products were collected using PT100 sensors (adhesive-type Class A, accuracy ± 0.15, temperature range −60~180 °C), which were connected to a paperless recorder (SIN-R9600, accuracy 2%, Hangzhou Lianmei Automation Technology Co., Ltd., Hangzhou, China). The air inlet velocity was measured at five different points using an anemometer (model Testo410i, range 0.4~30 m/s, accuracy ± (0.2 m/s + 2% of the measured value)), and the average value was taken.

The cold storage plates were made of 2 mm thick aluminum alloy (outer dimensions: 1 m × 0.04 m × 0.2 m) and are shown in Figure 3 with a 3 cm filling port on the top, which is used for the water injection (5 kg of water per plate), and a PT100 sensor was placed in the center of the cold storage plates to measure the PCM temperature.

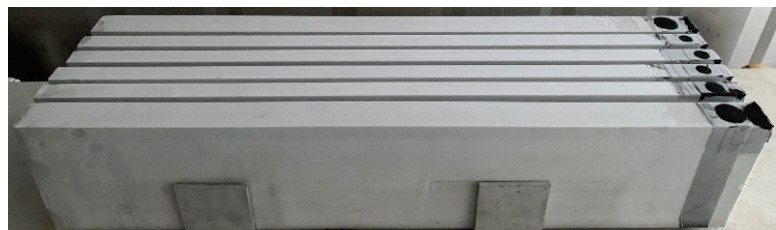

**Figure 3.** Cold storage plates.

To effectively utilize the results, the temperature at a representative point is often adopted to represent the overall temperature of the cold storage area. Therefore, only the temperatures of two pieces of the cold storage plates shown using the red points in Figure 2 were selected and averaged, and this value was used as the temperature representative point for the real-time status of all the cold storage plates. One and two PT100 sensors were arranged in the center and outer surface of each plate to measure its center and surface temperatures, respectively. The material physical parameters of the PCM (water) and the cold storage plates are shown in Table 3.

**Table 3.** Material physical parameters.

| Material | Phase Transition Temperature/°C | Latent Heat Value/(J·kg⁻¹) | Densities /(kg·m⁻³) | Heat Conduction /(W·(m·°C)⁻¹) | Specific Heat Capacity of the Solid State/(J·(kg·°C)⁻¹) | Specific Heat Capacity of a Liquid/(J·(kg·°C)⁻¹) |
|---|---|---|---|---|---|---|
| Water | 0 | 335,100 | 998 | 2.22 | 2050 | 4186 |
| Aluminum | - | - | 2700 | 237 | - | - |

The temperature-controlled container with a cold energy storage system was placed in an open and unobstructed outdoor environment for the test, and the air velocity of the fan was 4.4 m/s. Before the test, the cold storage plates were placed in a freezer ($-20\,°C$) to be frozen. Sixteen cold storage plates (containing a total of 80 kg of PCM) were moved to the cold storage area, and a basket (mesh structure with an outer dimension of 0.59 m × 0.415 m × 0.34 m) of 40 kg fresh navel oranges was placed in the fresh-keeping area. The test was started after the door of the container was closed, and the initial temperature at the center of the cold storage plates was $-8\,°C$. The temperature control range of the outlet temperature from the cold storage area was set at 5~10 °C, which was reported as the best fresh-keeping temperature for navel oranges [24], and the fan was turned off when the outlet temperature from the cold storage area dropped to 5 °C and was turned on when it rose to 10 °C. The test was completed when the value of the temperature at the center of the cold storage plates (PCM) rose to 10 °C.

**The temperature at the air inlet and outlet:** 2 PT100 sensors were placed at the air inlet and outlet (one near the outlet fan and one near the inlet fan) to measure the air temperature at the air inlet and outlet.

**Ambient temperature in the fresh-keeping area:** 2 PT100 sensors were placed in the fresh-keeping area to measure the ambient air temperature.

**Fresh products temperature:** 2 PT100 sensors were placed at the center of the navel oranges to measure their temperature.

**Surface temperature of cold storage plates:** 2 PT100 sensors were placed on the outer surface of the cold storage plates.

**Center temperature of cold storage plates:** 2 PT100 sensors were placed at the center of the cold storage plates.

The distribution of each temperature monitoring point is shown in Figure 4, and the average of the above sensors was taken as the final value.

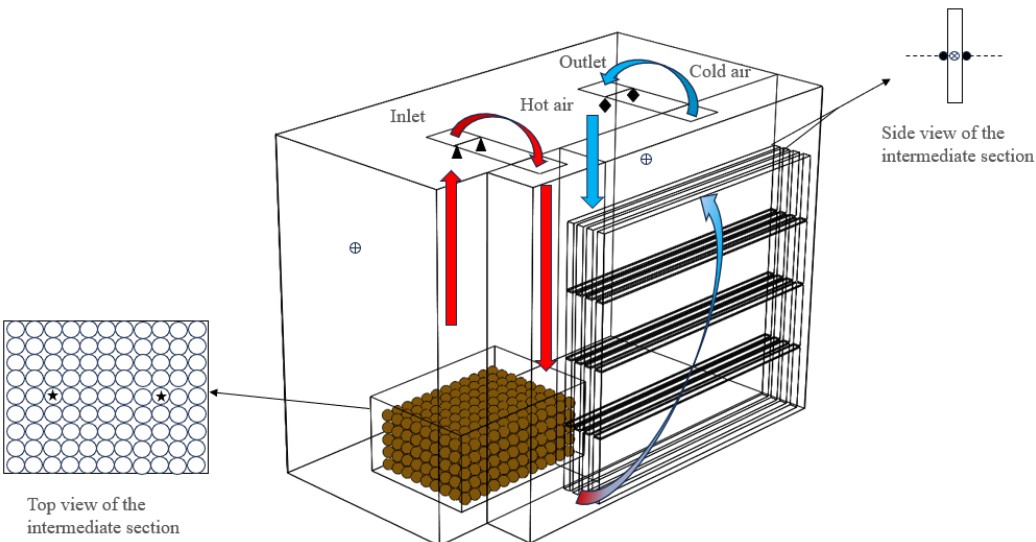

**Figure 4.** Distribution of temperature monitoring points: ★ —temperature monitoring points for fresh products; ▲ —temperature monitoring points for the inlet; ◆ —temperature monitoring points for the outlet; ● —temperature monitoring points for the surface of the cold storage plate; ⊗ —temperature monitoring points for the center of the cold storage plate; ⊕ —temperature monitoring points for the fresh-keeping area.

## 3. Physical Modelling of Heat Transfer

In order to verify the accuracy of the LSTM model, the prediction results of the LSTM model were compared with the physical model. This physical model is a differential format based on mathematical derivation with heat balance theory [25], and the accuracy of this model depends on the variation of heat flow inside and outside the container, as shown in Figure 5.

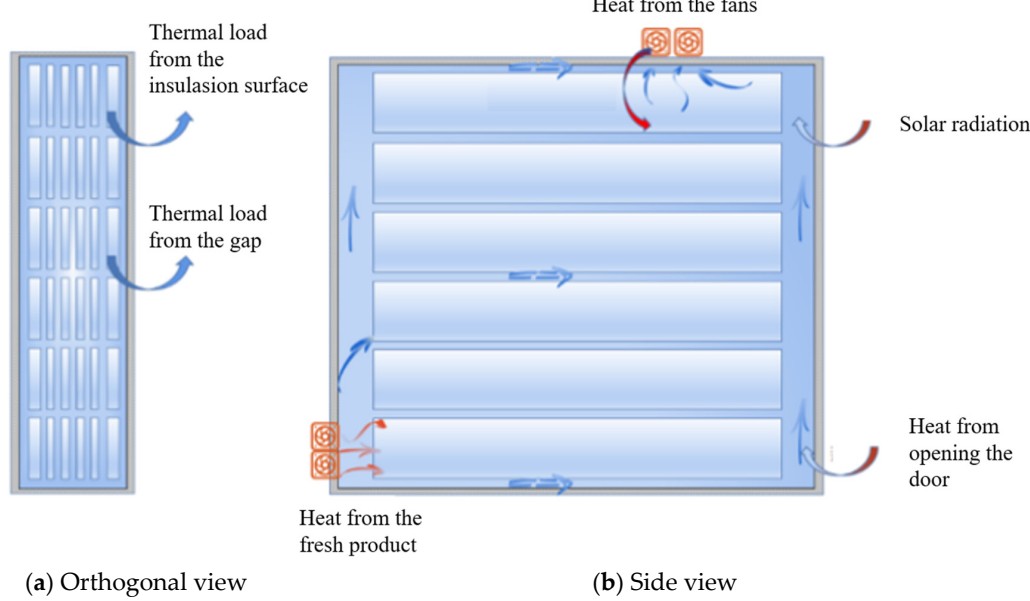

(**a**) Orthogonal view          (**b**) Side view

**Figure 5.** Analysis of heat transfer in a temperature-controlled container with a cold storage system.

The amount of instant cold released from the cold storage plates is represented by the following equation:

$$Q_g = AK\,(T_a - T_s) \tag{1}$$

where $Q_g$ is the instant cold released from the cold storage plates, W; $A$ is the heat transfer area on the cold storage plates, $m^2$; $K$ is the surface convection heat transfer coefficient of the cold storage plates, $W/(m^2 \cdot °C)$; $T_a$ is the average temperature in the cold storage area, °C, and is equal to the average of the inlet and outlet temperatures; and $T_S$ is the surface temperature of the cold storage plates, °C.

The heat transfer is driven by the temperature difference between the inside and outside of the insulation surface and can be represented by the following equations [26,27]:

$$Q_2 = A_w k_w \left(T_w - T_n\right) \tag{2}$$

$$k_w = \frac{1}{R_w} \tag{3}$$

$$R_w = \frac{1}{a_1} + \frac{1}{a_2} + \frac{\sum x_i}{\sum \lambda_i} \tag{4}$$

$$a = 1.1634 \left(4 + 12\sqrt{v}\right) \tag{5}$$

$$A_w = \sqrt{A_1 A_2}. \tag{6}$$

In the formula, $Q_2$ is the heat transferred due to the temperature difference between the inside and outside of the insulation surface, W; $k_w$ is the equivalent transfer coefficient for the whole container, $W/(m^2 \cdot °C)$; $T_w$ is the temperature outside the container, °C; $T_n$ is the temperature inside the container, °C; $A_w$ is the heat transfer area of the container, $m^2$; $R_w$ is the thermal resistance of the container, $(m^2 \cdot °C)/W$; $a_1$ is the heat transfer coefficient of the outside surface of the container, $W/(m^2 \cdot °C)$; $a_2$ is the heat transfer coefficient of the inner surface of the container, $W/(m^2 \cdot °C)$; $x_i$ is the thickness of each layer of insulation material, m; $\lambda_i$ is the thermal conductivity of each layer of insulation material, $W/(m^2 \cdot °C)$; $a$ is the coefficient of convective heat transfer between the insulation surface and the air, $W/(m^2 \cdot °C)$; $v$ is the speed of air flow in the container, m/s; $A_1$ is the total external surface area of the temperature-controlled container, $m^2$; and $A_2$ is the total internal surface area of the container, $m^2$.

$$Q_3 = f Q_2 \tag{7}$$

$$Q_4 = A_3 k_w \left(T_r - T_w\right) \frac{\tau_r}{24} \tag{8}$$

$$Q_5 = p\psi \frac{\varepsilon}{24} \tag{9}$$

In the formula, $Q_3$ is the thermal load from the gap leakage, W; $f$ is the air leakage coefficient of the container and can be between 0.1 and 0.2 based on the airtightness of the container [28], 0.1; $Q_4$ is the thermal load for solar radiation, W; $A_3$ is the area exposed to solar radiation (generally, between 30% and 50% of the total area is chosen; we chose 50%), $m^2$; $T_r$ is the temperature of the surface exposed to solar radiation ($T_r = T_w + 20$), °C; $\tau_f$ is the time of solar radiation per day and night, h; $Q_5$ is the heat from the fan operation, W; $p$ is the rated power of the fan, W; $\psi$ is the coefficient of thermal conversion, 1; and $\varepsilon$ is the coefficient of fan operation time [29], 19 h.

$$Q_6 = m_i q_i \frac{\tau_i}{24} \tag{10}$$

$$Q_7 = c_c m_c \left(T_c - T_d\right) / t_1 \tag{11}$$

where $Q_6$ is the respiratory heat of the navel oranges in the container, W; $m_{ii}$ is the mass of the navel oranges in the container, kg; $q_i$ is the respiratory heat flux of the navel oranges in the container [30], 0.15 W/kg; $\tau_i$ is the time for the navel oranges to emit respiratory heat in the container [31], 24 h; $Q_7$ is the thermal consumption of the navel oranges in the container, W; $c_c$ is the specific heat capacity of the navel oranges, 3.24 kJ/(kg·°C); $m_c$ is the

mass of the navel oranges, kg; $T_c$ is the initial temperature of the oranges, °C; $T_d$ is the temperature of the navel oranges at the end of the test, °C; and $t_1$ is the time in whole tests.

$$Q_8 = n\left(Q_2 + Q_3 + Q_4\right) \tag{12}$$

where $Q_8$ is the thermal load of the opening and closing of the container door, W; $n$ is the frequency coefficient of the opening of the door, 0.5 (open the door 0 times for 0.25; open the door 1 to 5 times for 0.5; open the door 6 to 10 times for 0.75; and open the door 11 to 15 times for 1).

The heat balance dynamic equation for the air-cooling process in the holding tank is shown below:

$$c_a m_a \frac{dT_n}{dt} = KA\left(T_a - T_s\right) - Q_2 - Q_3 - Q_4 - Q_5 - Q_6 - Q_8 - K_2 A_4\left(T_n - T_c\right) \tag{13}$$

where $c_a$ is the specific heat capacity of air, J/(kg·°C); $m_a$ is the mass of air, kg; $t$ is the time, s; $K_2$ is the heat transfer coefficient between the navel oranges and the air, W/(m²·°C); and $A_4$ is the heat transfer area between the navel oranges and the air [32], m².

The backward difference method, which specifies a 1 s time interval, was adopted [25] and then solved to show the temperature variation in the fresh-keeping area.

$$T_n = \frac{c_a m_a T_n\left(t-1\right) + KAT_s\left(t-1\right) + (1+f)A_w k_w T_w\left(t-1\right) + Q_4 + Q_5 + Q_6 + Q_8 + K_2 A_4 T_c\left(t-1\right)}{c_a m_a + KA + (1+f)A_w k_w + K_2 A_4} \tag{14}$$

## 4. LSTM Neural Network Model

### 4.1. Principle of LSTM Model

A typical LSTM cell is shown in Figure 6, where $\otimes$ and $\oplus$ denote multiplication and addition, respectively, by bits, $x_t$ is the input parameter at moment t, $c_t$ is the output parameter from the memory cell at moment t, $h_t$ is the output parameter of the control cell at moment t, $c_{t-1}$ is the memorized information of the previous moment, $f_t$ is the probability that $c_{t-}$ is forgotten, $i_t$ is the probability that the candidate valuegenerated from the activation function tanh $x$enters the memory cell, $\hat{c}_t$ is the memory candidate value after the current data were transformed, and $o_t$ is the probability that the current data become the output data. The output parameter $h_t$ is not only related to the input $x_t$ but also to the previously remembered information $c_{t-1}$, and the new input $x_t$ will also prompt the memory cells to generate updates that are carried forward [20]. The output of the implicit layer is mapped using the function of the output layer to obtain the predicted value of the temperature.

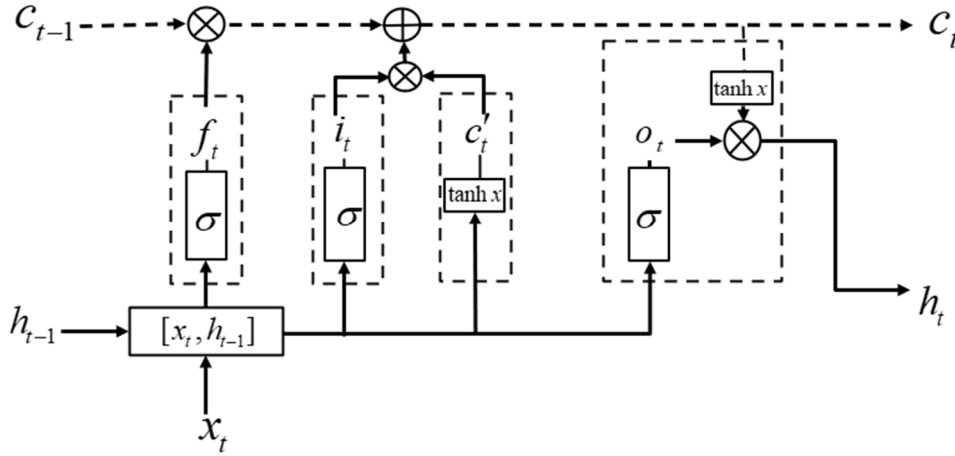

**Figure 6.** Schematic diagram of LSTM.

The calculation formulas for the LSTM model are shown as follows:

$$
\begin{cases}
f_t = \sigma\left[W_f \cdot (h_{t-1}, x_t) + b_f\right] \\[2mm]
i_t = \sigma[W_i \cdot (h_{t-1}, x_t) + b_i] \\[2mm]
o_t = \sigma[W_o \cdot (h_{t-1}, x_t) + b_o] \\[2mm]
h_t = o_t \cdot \tanh(c_t) \\[2mm]
c_t = F_t \cdot c_{t-1} + i_t \cdot c'_t \\[2mm]
c'_t = \tanh[W_c \cdot (h_{t-1}, x_t) + b_c]
\end{cases}
\tag{15}
$$

where $W_f$, $W_i$, $W_o$, and $W_c$ are the parameter matrices to be trained and $b_f$, $b_i$, $b_o$ and $b_c$ are the bias terms to be trained.

### 4.2. Predictive Model Construction and Parameter Settings

On the basis of the historical temperature data, the LSTM was used to predict the temperature inside the temperature-controlled container with a cold energy storage system. Initially, 62.95 h (226,604 s) was used as the sample for the dataset, the initial 80% of the dataset was used as the training set for training and model optimization, and the spare 20% of the data were used as the test set to evaluate the generalization and prediction accuracy of the LSTM model; the normalization process was carried out before inputting the data into the model to achieve a better prediction by continuously fine-tuning the parameters and decreasing the error [33].

The input layer contained 181,267 neurons, the output layer contained 45,317 neurons, and the hidden layer contained 2 LSTM layers with 32 neurons in each layer. The Adam optimization algorithm with a mean square error loss function (MSE) was adopted, the learning rate was set to 0.01, the sample was trained 100 times, the batch size (the number of samples selected for one training) was 19, and the temperature data from the initial 19 s were used in each step to predict the temperature data for the next 1 s [34].

### 4.3. Model Assessment

In order to evaluate the prediction accuracy of the model and facilitate the comparison and optimization of the model, the root mean square error (RMSE), mean absolute error (MAE), mean percentage error (MAPE) [35], and the coefficient of determination ($R^2$) were selected as the evaluation indexes to evaluate the accuracy of the model prediction [36]. The smaller the values of the RMSE, MAE, and MAPE, the smaller the deviation of the model prediction value compared to the real value; the closer $R^2$ is to 1, the greater the goodness-of-fit, which corresponds to a better model prediction performance. The specific calculation formulas are as follows:

$$
\text{RMSE} = \sqrt{\frac{1}{N}\sum_{i=1}^{N}(\hat{y}_i - y_i)^2}
\tag{16}
$$

$$
\text{MAE} = \frac{1}{N}\sum_{i=1}^{N}|y_i - \bar{y}_i|
\tag{17}
$$

$$
\text{MAPE} = \frac{1}{N}\sum_{i=1}^{N}\frac{|y_i - \bar{y}_i|}{y_i}
\tag{18}
$$

$$R^2 = 1 - \frac{\sum_{i=1}^{N} |y_i - \overline{y}_i|^2}{\sum_{i=1}^{N} |\overline{y}_i - \overline{y}_i|^2} \tag{19}$$

where $N$ is the number of data points in the dataset, $y_i$ is the actual value of temperature, $\hat{y}_i$ is the predicted value of temperature, and $\overline{y}_i$ is the mean of the true values.

## 5. Results and Analyses

### 5.1. Experimental Results

The average outdoor ambient temperature during the test was 24.5 °C. The temperature variation with time at each monitoring point is shown in Figure 7. The test was conducted for 62.95 h. At the beginning, due to the large temperature difference between the initial ambient environment and the cold storage plate, the temperature of the inlet, outlet, fresh-keeping area, and fresh products dropped rapidly under the work of the cold storage system, and it took 12.51 h for the navel oranges to be cooled down from the initial temperature to the target temperature of 10 °C. This also means that the fresh product was maintained at a lower temperature for the majority of the experiment. The temperature-controlled interval of the outlet temperature (5~10 °C) lasted 50.44 h. The average temperature of the fresh-keeping area within the temperature-controlled interval was 8.5 °C, and the average temperature of the products was 8.3 °C, which suggests that the retention and release rates of cold energy from the PCM allowed the temperature of the holding area to be maintained in a suitable range for a long period. The center temperature of the cold storage plates was maintained near the theoretical temperature of 0 °C (−2~2 °C interval) for 46.07 h, which accounted for 73.2% of the total test duration, reflecting the fact that the cold energy is mainly stored as the latent heat of the phase change material [37]. After 46.07 h, the temperature at the center of the plate rose faster because the PCM had ended the phase transition process, and the value of the difference between its temperature and the ambient temperature was greater than 8 °C, which formed a higher heat transfer rate. The temperature curves of the other monitoring points changed smoothly because the temperature difference between them and the environment was relatively small, and there was not a high rate of heat exchange. The fan was activated 88 times (9.06–45.71 h) during the test. The slight discrepancy observed between the center and surface temperatures of the cold storage plates during the test could be attributed to the uniform temperature distribution of the phase change material (PCM) during the phase transition, resulting in a surface temperature measurement that closely approximates the center temperature.

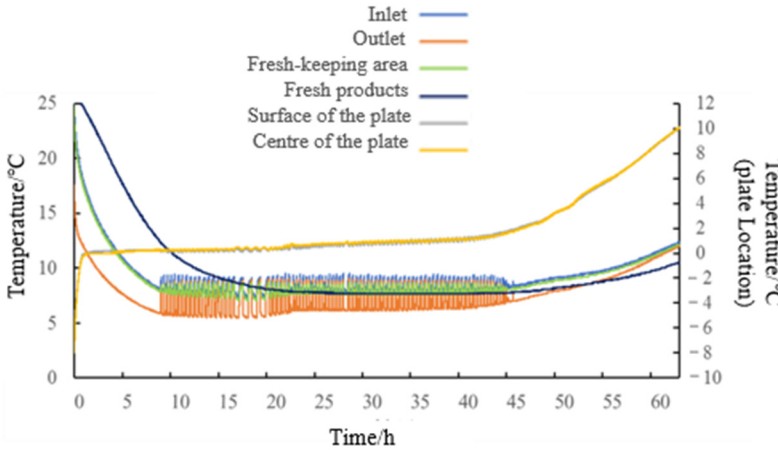

**Figure 7.** Temperature change at each monitoring point.

### 5.2. LSTM-Based Benchmark Model Results

The LSTM neural network model was built and trained on the PyTorch 1.3 platform to predict the temperature in the fresh-keeping area. The prediction results and training results of the LSTM-based temperature prediction model are shown in Figures 8 and 9, respectively.

The training results show that the LSTM model was able to accurately track the changes between the upper and lower limits of the temperature and predict the future temperature; on the other hand, the predicted values were in agreement with the actual values.

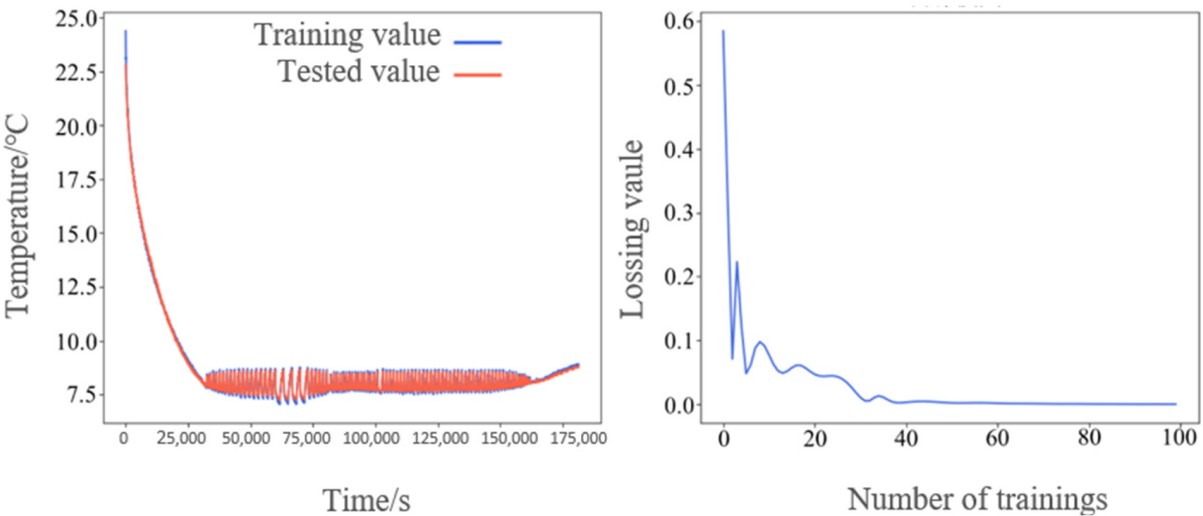

**Figure 8.** Training set results.

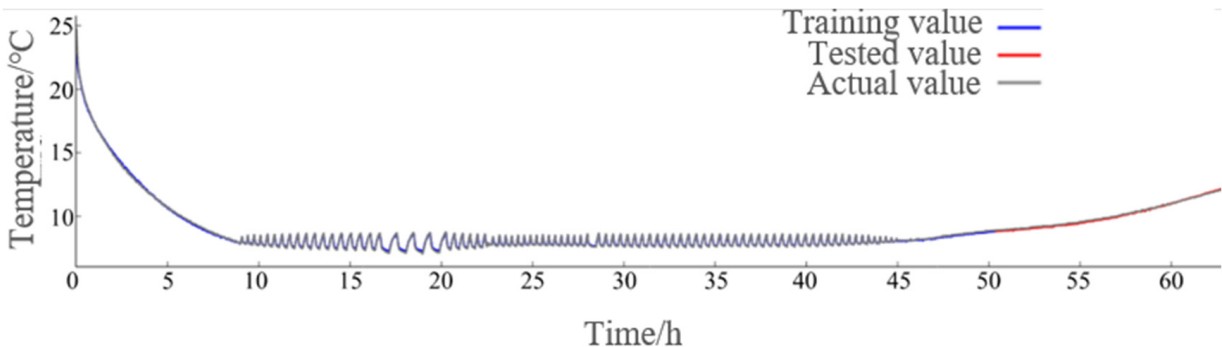

**Figure 9.** Test set temperature prediction result.

*5.3. Comparison of Predicted Results*

As shown in Figure 10 and Table 4, the RMSM, MAE, MAPE, and $R^2$ of the temperature prediction results based on the LSTM model were 0.105, 0.103, 0.010, and 0.988, respectively. In addition, the RMSM, MAE, MAPE, and $R^2$ of the temperature prediction results based on the physical model were 3.857, 2.554, 0.258 and 0.346, respectively. The accuracy of the temperature prediction model based on the LSTM was significantly higher than that of the physical model; the RMSE, MAE, and MAPE were reduced by 3.752, 2.451, and 0.248, respectively, the $R^2$ improved by 0.642, and the relative error in predicting the moment of CCB (cold chain breaking, which means the value of temperature rises to an unsuitable range for freshness) was reduced by 3.92%. The predicted temperature trend based on the LSTM model closely matched the actual temperature changes, indicating that the LSTM model is more adept at forecasting time-series ambient temperature fluctuations than the physical model of the temperature-controlled container. In addition, the physical model describes changes in heat transfer, but hardly for more in-depth explanations, such as the effect on temperature of some factors like the direction of airflow in the interior and the complex changes in the rate of cold energy release during phase transitions, which may be some of the reasons for the difference between the actual value and the predicted value.

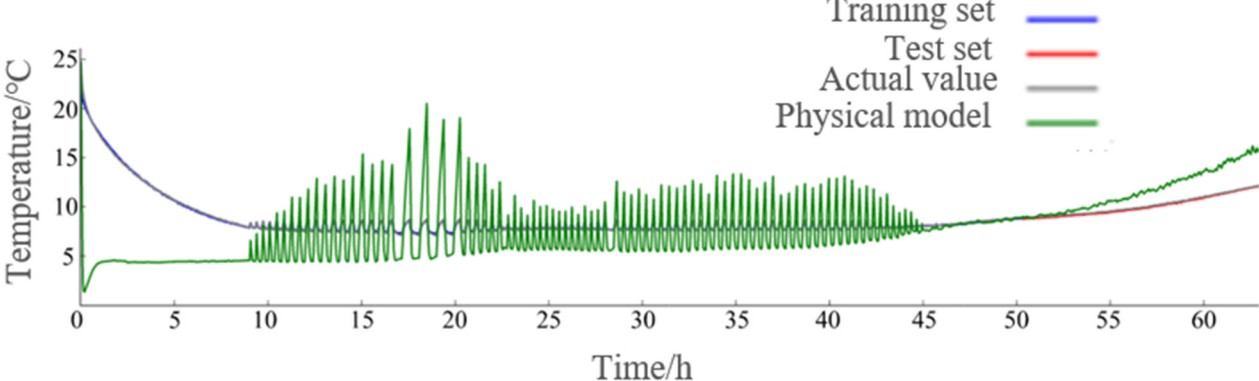

**Figure 10.** Comparison of the predictions of the two models.

**Table 4.** Comparison of the predictions of the two models.

| Model | RMSE | MAE | MAPE | $R^2$ | Time for Temperature to Reach 10 °C/h | | Relative Error/% |
|---|---|---|---|---|---|---|---|
| | | | | | Predicted | Actual | |
| LSTM model | 0.105 | 0.103 | 0.010 | 0.988 | 57.26 | 56.74 | 0.92 |
| Physical model | 3.857 | 2.554 | 0.258 | 0.346 | 53.99 | 56.74 | 4.84 |

## 6. Model Optimization

To maintain the model's predictive accuracy while enhancing its practicality, the number and structure of the inputs and outputs were optimized. This not only allows for more precise future predictions of the CCB but also reduces the model's training time and improves its generalizability [38]. The initial 80%, 50%, 20%, and 10% of the samples were configured as training sets, and the corresponding training and prediction results are shown in Figure 11 and Table 5. It can be seen that the accuracy of the prediction results of the different methods with the training set of the initial 80%, 50%, and 20% are similar, indicating that there is no need to have a training set consisting of too many samples if the characteristics of the time series are regular. The training time of the model decreases significantly with a reduction in samples in the training set, but the prediction accuracy, $R^2$, decreases significantly only when the number of samples in the training set is the initial 10%. Using the initial 20% of the historical temperature data to predict the future 80% of the temperature data is the most practical approach; the training time and $R^2$ are 932 s (15.5 min) and 0.997, respectively.

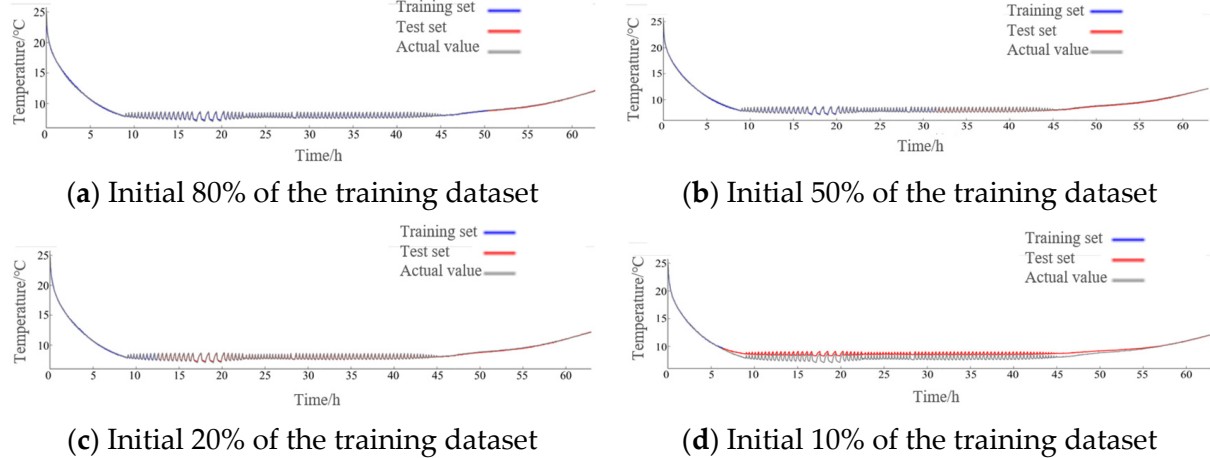

(**a**) Initial 80% of the training dataset

(**b**) Initial 50% of the training dataset

(**c**) Initial 20% of the training dataset

(**d**) Initial 10% of the training dataset

**Figure 11.** Comparison of prediction results for different training datasets.

**Table 5.** Comparison of prediction results for different training datasets.

| Number | Training Sample/% | Training Time/s | RMSE | MAE | MAPE | $R^2$ | Time for Temperature to Reach 10 °C/h | | Relative Error/% |
| --- | --- | --- | --- | --- | --- | --- | --- | --- | --- |
| | | | | | | | Predicted | Actual | |
| 1 | 80 | 2487 | 0.073 | 0.066 | 0.007 | 0.994 | 57.11 | 56.74 | 0.66 |
| 2 | 50 | 1740 | 0.102 | 0.087 | 0.010 | 0.993 | 57.26 | 56.74 | 0.93 |
| 3 | 20 | 932 | 0.059 | 0.051 | 0.006 | 0.997 | 57.11 | 56.74 | 0.66 |
| 4 | 10 | 515 | 0.651 | 0.580 | 0.073 | 0.631 | 56.62 | 56.74 | 0.20 |

## 7. Applications of the Model

The LSTM model was employed to accurately capture the temperature fluctuations at the cold energy storage source, the surface of the cold storage plates, the fresh products, and the air inlet and outlet of the container. It predicts the temperature changes at these locations, offers valuable insights into the timely replacement of cold storage plates, monitors the status of the CCB during transport, and ensures optimal temperatures are maintained in the fresh-keeping area.

The training set of the model was set to the initial 80% of the samples, and the value of the CCB temperature of the PCM and fresh products was set to 5 °C and 10 °C, respectively. The training and prediction results are shown in Figure 12a–e and Table 6. As shown in Figure 12, the predicted temperature variations are all in agreement with the actual values. As shown in Table 6, the relative errors in predicting the status of CCB in cold energy storage sources and fresh products are 1.05% and 0.45%, respectively, indicating that the LSTM model can accurately predict the status of CCB. In addition, the RMSE and MAE of the temperatures of the fresh products, inlet, outlet, and surface of cold storage plates are all below 1.000, the MAPE is below 0.1000, and the $R^2$ of the temperatures of the fresh products, inlet, and outlet are slightly higher, with values exceeding 0.9400, which indicates that the LSTM model accurately predicts the temperatures of the fresh products, inlet, outlet, and the surface of cold storage plates.

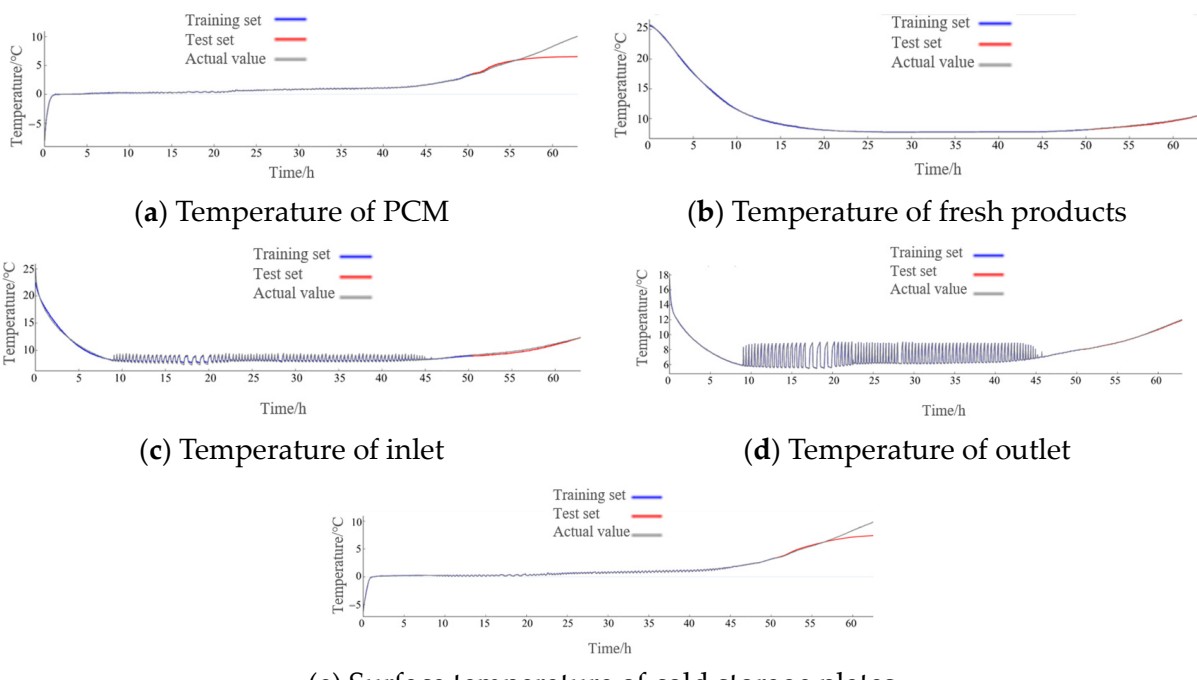

(**a**) Temperature of PCM

(**b**) Temperature of fresh products

(**c**) Temperature of inlet

(**d**) Temperature of outlet

(**e**) Surface temperature of cold storage plates

**Figure 12.** Training and prediction results.

**Table 6.** Training and prediction results.

| Elements | RMSE | MAE | MAPE | $R^2$ | Time to Reach the Moment of CCB/h | | Relative Error in CCB prediction/% |
|---|---|---|---|---|---|---|---|
| | | | | | Predicted | Actual | |
| PCM | 1.442 | 0.975 | 0.121 | 0.450 | 52.98 | 53.54 | 1.05 |
| Fresh products | 0.091 | 0.087 | 0.010 | 0.981 | 61.52 | 61.24 | 0.45 |
| Inlet | 0.223 | 0.212 | 0.021 | 0.944 | - | - | - |
| Outlet | 0.041 | 0.029 | 0.003 | 0.999 | - | - | - |
| Surface of cold storage plates | 0.991 | 0.640 | 0.079 | 0.746 | - | - | - |

One of the reasons for the low $R^2$ of the temperature prediction of the cold energy storage source and surface was that the temperature data in the training set were maintained near the phase change temperature of 0 °C for a long time, and the increasing trends were not obvious, resulting in the predicted values in the test set not rising as drastically as the actual values.

**8. Conclusions**

This study was based on the temperature monitoring data from a temperature-controlled container with a cold energy storage system. A long short-term memory network (LSTM) temperature prediction model was built and compared with the physical model. After that, the dataset for training was optimized to achieve a more accurate prediction of the temperature in the container, and the conclusions were drawn as follows.

Compared with the physical model, the LSTM model predicted the trend with higher accuracy, the RMSE, MAE, and MAPE were reduced by 3.752, 2.451, and 0.248, respectively, the $R^2$ improved by 0.642, and the relative error in predicting the status of the CCB (cold chain breaking) temperature was reduced by 3.92%.

The optimization results of the model dataset for the temperature prediction in the fresh-keeping environment showed that it is recommended to use the initial 20% of the historical temperature data to predict the future 80% of the temperature data, with a training elapsed time and $R^2$ of 15.5 min and 0.997, respectively.

The LSTM model was able to achieve accurate predictions of the status of CCB for PCM in the cold storage plates and the fresh products with a relative error of 1.05% and 0.45%, respectively.

The above shows that the temperature prediction of temperature-controlled containers with a cold storage system using an LSTM model has high accuracy. However, there are still points that need to be improved. The LSTM model is not adept at handling a long series of data; the prediction accuracy of the LSTM model is affected when the time demand for temperature prediction is long. In addition, if it is possible to select the training set separately according to the phase transition phase and the non-phase transition phase, then this will be beneficial in improving the accuracy of the temperature prediction and reduce the training time. Finally, it is worthwhile to test different neural network models for temperature prediction because more efficient neural networks perform better in applications.

**Author Contributions:** Conceptualization, J.G. and S.L.; methodology, S.L.; software, S.L.; validation, J.L.; formal analysis, J.G.; investigation, S.L., J.L. and D.L.; resources, W.Z. and J.G.; data curation, D.L.; writing—original draft preparation, D.L.; writing—review and editing, J.G.; visualization, D.L.; supervision, J.G.; project administration, W.Z.; funding acquisition, W.Z. All authors have read and agreed to the published version of the manuscript.

**Funding:** This research was funded by the Guangdong Province 2019 Provincial Agricultural Science and Technology Innovation and Extension Project, grant number 2023KJ101; the Innovative Team for Common Key Technology Research and Development of Agricultural Product Freshness and Logistics, grant number 2023KJ145; the National Science Foundation of China, grant numbers 31901736

and 31971806; the Maoming Laboratory Autonomous Scientific Research Project, grant number 2021ZZ003; the Natural Science Foundation of Guangdong Province, grant number 2020A1515010967; and topics of R&D in Key Areas of Guangzhou Municipality, grant number 2023B01J2001.

**Institutional Review Board Statement:** Not applicable.

**Informed Consent Statement:** Not applicable.

**Data Availability Statement:** The data presented in this study are available on request from the corresponding author. The data are not publicly available due to Policy and non-disclosure agreement.

**Acknowledgments:** The authors would like to express their appreciation for the valuable suggestions and support of Qunyu Shen and Xiaodan Zhang.

**Conflicts of Interest:** The authors declare no conflicts of interest.

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
