# Peer review of "Temperature Prediction of a Temperature-Controlled Container with Cold Energy Storage System Based on Long Short-Term Memory Neural Network"

_applsci, doi:10.3390/app14020854_

Round 1

Reviewer 1 Report

Comments and Suggestions for Authors

The paper deals with a very interesting topic, i.e. the temperature prediction in a Cold Storage Device, by means of LSTM Neural Network.

The figures and tables are clear and also le mathematical model is clearly explained. The conclusions well summarize the results of the paper.

Even if the scientific soundness is good, I strongly suggest some improvements:

1)      Please, improve the quality of the figures (for example Fig. 6)

2)      Please, improve the quality of the equations (for example eq. 12).

3)      As regards the figures that show the performances of the NN in the temperature prediction (see Figs. 9-10) , it will be useful to the reader if the authors would report also the difference between the temperature of the actual case and the predicted one.

4)      Why didn’t the authors use other NN such as the Fourier Decomposition State-Space Gated Recurrent Unit?

5)      Did the authors test the prediction performances also on other measured datasets?

Reviewer 2 Report

Comments and Suggestions for Authors

Dear,

The current article provides a comprehensive overview of control methods for dynamic systems, addressing crucial aspects related to the quality of figures, equations, the state of the art, technical innovation, experimental details, and presented results. However, there are some points to be improved to strengthen the scientific contribution of the work.

1. Quality of Figures: The figures lack visual quality, requiring enhancement for clarity and visual appeal. The use of specialized software is recommended to improve resolution and formatting, ensuring precise and understandable data representation.

2. Quality of Equations: Equations also need significant improvement in terms of clarity and presentation. Revision using mathematical text editors is suggested to ensure consistent and easily understandable notation. Proper equation formatting is crucial for a precise understanding of the discussed methods.

3. State of the Art:  While emphasizing the importance of detailing the state of the art in control methods, a deeper analysis of the current landscape, including discussions on associated errors and response times, is needed to strengthen this section.

4. Technical Innovation: It is essential to clearly highlight the innovation or addition brought by the proposed technique, articulating the specific contribution explicitly and demonstrating how it differs or improves upon existing techniques.

5. Experimental Detailing and Results: The experimental section and presented results require more detailed information. Specifics on experimental procedures should be provided for transparency and replicability. Additionally, results should be presented clearly with an in-depth analysis of observed trends and their implications.

6. Similarity with Documents: The report mentions a high similarity index with attached digital documents. To maintain academic integrity, this issue must be addressed transparently. Proper citations and references should be incorporated if applicable, ensuring the originality and credibility of the work.

In conclusion, the review and enhancement of these points will significantly contribute to the quality and impact of the article. Careful attention to these aspects will ensure a valuable contribution to the field, strengthening the credibility and relevance of the presented work.

Reviewer 3 Report

Comments and Suggestions for Authors

This manuscript discusses about using the LSTM model to predict temperature of the cold storage applications. Therefore, it matches with the scope of Applied Sciences.

However, some concerns and questions still need to be addressed:

Major:

1.) Page 1-2: it is not well and clearly stated about the questions resolved by this work, could the authors please clearly state the limitations/questions of this area before this study, and the novelty of this work ?

2.) Page 2, section 2: could the authors specify some detailed information on the contained in an quantitative way, for example, what are the size and speed of the fan, what are the materials used to construct the container ? 

3.) Page 9, line 289: could the authors comment on why the experiments were conducted for 62.95 h ? Would integer times of an entire day be more reasonable ?

4.) Page 13, the conclusion section: could the authors comment on the limitations or the places need to be improved for the LSTM model investigated here for temperature prediction ?

Minor:

a.) Some places need to be corrected for grammar, for example, Page 1, line 25 “The results of the study shows that….”, Page2, line 51 “There are a large number of literatures on such problems.”.

Based on the current status of the manuscript, I recommend a minor revision.

Comments on the Quality of English Language

The English quality is a concern, there are places not right in grammar and not clearly stated.

Reviewer 4 Report

Comments and Suggestions for Authors

The paper deals with a very interesting topic: the temperature prediction of a container with Cold Energy Storage System, by using the LSTM Neural Network.

The paper is well written and the model is well presented. The conclusions well resume the paper results.

The figures are clear, but:

1)    Please, pay attention to the format of the equations (see for example eqs. 1, 2, 7, 9).

2)    Please, improve the quality of the figures (see for example Fig. 6).

Finally, a question on the adopted model: Why did not the authors use models as the Fourier Decomposition State-Space Gated Recurrent Unit (FD-SS-GRU)?

Round 2

Reviewer 2 Report

Comments and Suggestions for Authors

The authors responded to my requests and I am satisfied.

Just as a request, I suggest changing the conclusions to paragraph format.

Congratulations
